# Variance in Stallion Semen Quality among Equestrian Sporting Disciplines and Competition Levels

**DOI:** 10.3390/ani9080485

**Published:** 2019-07-25

**Authors:** Megan Wilson, Jess Williams, V. Tamara Montrose, Jane Williams

**Affiliations:** Equine and Animal Departments, Hartpury University, Gloucester GL19 3BE, UK

**Keywords:** equine breeding, reproduction, semen quality, stallion

## Abstract

**Simple Summary:**

Despite the increased demand for breeding stallions to be performing at elite levels of competition, little research has been conducted into how equestrian disciplines and competition level affect seminal quality. Using statistical analysis, we found that competing stallions have lower quality semen than non-competing stallions and competitive activity may have a greater negative impact on spermatogenesis than age. In addition, dressage stallions recorded improved semen characteristics when compared to show jumping and eventing stallions. Stallions at elite levels of competition recorded higher total sperm count and progressive motility compared to those competing at lower levels of competition. We suggest that appropriate management of the competition stallion may reduce the negative impacts which exercise can induce on semen quality.

**Abstract:**

Most stallions within breeding programmes are expected to breed and compete concurrently. The exercising of stallions with regards to training regimes during the breeding season is a controversial subject. Daily exercise at low intensities is important for the mental and reproductive well-being of the stallion, however higher intensities of exercise, as seen in competing stallions, may have detrimental effects on seminal quality. To calculate if competition does affect semen quality, this study investigated the effect that equestrian discipline and timing of competition had on a range of stallion semen characteristics. This was a retrospective study that evaluated the seminal data of 1130 stallion semen collections from two UK based stud farms between 2009 and 2016. Competing stallion semen quality was significantly lower with regards to concentration (*p* < 0.05) and progressive motility (*p* < 0.05) than non-competing stallions. Semen volume was higher in competing stallions (*p* < 0.05) than non-competing stallions. There was a significant difference in seminal attributes among disciplines and competition levels (*p* < 0.05). The difference in semen quality among competing and non-competing stallions, as well as the difference among disciplines suggests endocrinological and physiological changes occur in relation to training intensity and competition.

## 1. Introduction

The aim within British sport horse breeding is to produce offspring that will, in the future, perform successfully within international competitions [1,2]. This need for constant genetic progression means that a successful career in competition is considered vital to increase a horse’s potential as a valuable breeding stallion [3]. Consequently, many sports horse stallions have dual careers and often need to fit breeding schedules around competition dates.

Physiological and endocrinological responses to exercise can affect spermatogenesis [4,5,6] and therefore produce fluctuations in seminal characteristics [7,8,9,10]. Research in humans [11,12] has found that different sporting modalities can influence semen quality through physiological and endocrinological changes associated with intensity, frequency, duration and type of exercise [11,12].

Even though exercise modality and intensity has been frequently researched within human science, this is not the case for the equine industry. Exercise within different equestrian disciplines results in varied physiological and endocrinological responses similar to the changes observed in humans. It is therefore likely that changes also occur within equine spermatogenesis and semen quality in response to exercise loads and training regimes [13]. Studies investigating the effects of exercise upon semen quality in the horse have produced conflicting results to date: some papers state that exercise has no impact on stallions’ semen quality [7,10], disagreeing with research that has concluded that exercise has a significant effect on stallion semen characteristics [8,9,14].

Factors that affect spermatogenesis, such as thermal stress, hormones and oxidative stress, have the potential to negatively impact stallions’ fertility through detrimental effects on semen characteristics. However, there is little research evaluating the effects of exercise in competitively active stallions, despite the increasing popularity of dual purpose (breeding and competition) stallions in the industry. To calculate if competition affects semen quality, this study investigated the effect that equestrian discipline, competition level and timing of competition have on a range of stallion semen characteristics (total volume, gel-free volume, sperm concentration, sperm progressive motility, total sperm count and total number of progressively motile sperm).

## 2. Materials and Methods

Retrospective data for six semen performance determinants were collected from two UK based stud farms for a seven-year period (2009–2016). Data were collected for 142 resident sports horse stallions aged between two and twenty-five years (9.21 ± 4.69 years) used within artificial insemination breeding programmes. Stallion breed was controlled to include only warmbloods and competition level was controlled in accordance with previous research [15,16,17,18]. To be included in the study, stallions were required to be warmbloods, and to participate in show jumping, dressage or eventing, or be non-competing. Stallions from any other breeds, any other discipline and/or participating in more than one discipline were excluded from the sample. Competition levels were split into three categories: lower levels (show jumping: unaffiliated, novice, discovery, newcomers; dressage: unaffiliated, intro, prelim, novice, elementary; eventing: unaffiliated, BE80–BE100); higher levels (show jumping: 1.20–1.35 m; dressage: medium, advanced medium, advanced; eventing: novice, intermediate, 1 star (star levels (1–5) describe Federation International Equestre (FEI) international level Eventing competitions), 2 star; and elite levels (show jumping: ≥1.40 m+; dressage; prix saint George, intermediate I, intermediate II, grand prix; eventing: advanced, 3 star, 4 star).

For each stallion, age, breed, discipline, competition level and dates competed were recorded alongside key seminal characteristics identified from previous research [15]. Volume of semen sample (mL), volume of gel-free sample (mL), sperm progressive motility (percentage), sperm concentration (million/mL), total sperm count (billion), total progressively motile sperm (billion) and collection date were recorded. All samples were collected and assessed by Department for Environment, Food and Rural Affairs (DEFRA) qualified Artificial Insemination (AI) technicians and both stud farms used the same techniques for semen evaluation. Sperm concentration was analysed using an automated sperm cell counter (NucleoCounter SP-100) and motility was assessed via microscopic evaluation.

In total, 1601 semen collections were included for analysis, 400 from non-competing stallions and 1201 from competing stallions (show jumping, *n* = 422; dressage, *n* = 397; eventing, *n* = 382). The majority of stallions included (93.6%) had multiple sample collections within the data, however as environmental conditions, management and competition schedules have previously been shown to impact semen quality [19,20], each collection was treated as a unique and individual data point.

Ethical approval for the study was granted by the Hartpury Ethics Committee. The ethics code for this project was: ETHICS2018-57 awarded from the Hartpury Ethics Committee. Only DEFRA approved stud farms were used within this research, anonymity was maintained throughout and data were collected and maintained in accordance with the Data Protection Act 1988 [21].

### 2.1. Data Analysis

#### 2.1.1. Descriptive Analysis

Descriptive analysis of data was performed to establish mean ± standard deviation (sd), medians, ranges and interquartile ranges for the semen characteristics recorded. The frequency of collections for age, discipline and competition level were also recorded. Differences in semen collection technique (number of mounts, teasing time, and the artificial vagina used) were not factored into the model as these data were unknown.

#### 2.1.2. Effects of Discipline, Competition Level and Age on Semen Characteristics

Data were non-parametric, therefore a series of Kruskal–Wallis analyses were conducted to determine whether stallion discipline, stallion competitive level or stallion age significantly affected seminal characteristics (total volume, gel-free volume, sperm concentration, sperm progressive motility, total sperm count and total number of progressively motile sperm). Where these tests found significant differences, post-hoc analysis with Mann–Whitney U tests determined where differences occurred among groups. A Bonferroni adjustment was applied to control for type I errors, resulting in a modified alpha value of *p* < 0.008 for the discipline and competition level, and *p* < 0.005 for the age post-hoc analyses.

A two-way ANOVA with log-transformation assessed if interactions occurred between age and discipline, and age and competition level for the semen characteristics assessed. Where significant interactions occurred, post-hoc paired T-tests with pooled standard deviations were carried out (alpha: *p* < 0.05).

#### 2.1.3. Multivariable Modelling: Effect of Related Factors on Semen Characteristics

Univariate logistic regression informed multivariate model building. Individual variables were tested against the dichotomous outcome of being above or below the industry standard measure for each of the semen characteristics evaluated (gel-free volume; >40 mL: sperm concentration; 100–450 × 10^6^ sperm/mL: progressive motility; >40%: total sperm count; >6 × 10^9^ sperm: total number of progressively motile sperm; >2.4 × 10^9^ sperm) [22]. Variables with an alpha value of *p* < 0.10 were considered for use in building the multivariable models [23]. In addition to significant variables (*p* < 0.05), timing between competition and semen collection, discipline and all semen characteristics were considered for inclusion in all models based upon previous research [6]. Age was also included in all models as previous studies highlight this as a biologically plausible factor related to semen quality [15,16]. For the purpose of this research, 12 predictive multivariable binary logistic regression models were produced; data from all stallions (Model A 1–6) and data from competing stallions only (Model B 1–6) (Appendix A). Models were fitted using a backward stepwise method that excluded variables with a likelihood ratio test significance of *p* < 0.05. For each step of the model building process, model fit was evaluated using an Omnibus test, Nagelkerke’s R2 and Hosmer–Lemeshow goodness of fit tests (*p* < 0.05). The predictive abilities of the final models were investigated using receiver operating characteristic (ROC) curve analysis [24].

Statistical analyses of the results were performed using SPSS (Version 23.0) and R (Version 3.3.3).

## 3. Results

### 3.1. Descriptive Results

Stallion data included 1601 semen collections, with similar numbers recorded for each discipline: 24% (*n* = 382) in eventing, 26% (*n* = 422) in show jumping, 25% (*n* = 397) in dressage and 25% (*n* = 400) in non-competing stallions. The majority of stallions were aged between five and nine years and the level horses were competing at varied across the discipline (Figure 1).

### 3.2. Difference in Semen Characteristics among Age Categories

Differences were found in the average and median values for the semen factors investigated which related to the age of the stallions that semen was collected from (Table 1). Kruskal–Wallis analyses found significant differences (*p* < 0.0001) for all semen characteristics across the stallion age categories investigated. Post-hoc tests identified that multiple significant differences in semen characteristics existed among stallions in the different age groups (*p* < 0.005; Table 2). Younger stallions aged 2–4 years had lower total volumes (TV) and gel-free volumes (GFV) than stallions aged 5–9 years (24% decrease in TV; 21% decrease in GFV, *p* < 0.0001) and 10–14 years (27% decrease in TV and GFV, *p* < 0.0001). However, the younger age group had a 30% increase in sperm concentration compared to stallions aged 10–14 years (*p* < 0.0001). Stallions aged over 20 years also consistently recorded higher values for sperm progressive motility (19% increase to 2–4 years; 19% increase to 5–9 years; 22% increase to 10–14 years; 24% increase to 15–19 years), sperm concentration (32% increase to 2–4 years; 38% increase to 5–9 years; 47% increase to 10–14 years; 35% increase to 15–19 years), total sperm count (25% increase to 2–4 years; 11% increase to 5–9 years; 29% increase to 10–14 years; 23% increase to 15–19 years) and total progressively motile sperm (35% increase to 2–4 years; 23% increase to 5–9 years; 47% increase to 10–14 years; 43% increase to 15–19 years) than younger horses (*p* < 0.0001). Interestingly, significantly reduced values for sperm progressive motility and sperm concentration occurred between stallions aged 10–14 years, who recorded 4% and 19% lower values respectively (*p* < 0.004) compared to stallions aged 5–9 years, and 19% reduced sperm concentrations than stallions aged 15–19 years (*p* < 0.002). Differences in total volume and gel-free volume were also found between stallions aged 5–9 years and 10–14 years to stallions aged 20 years and over. Total volume was reduced in the older stallions by 34% to 5–9 year olds and 38% to 10–14 year olds (*p* < 0.0001). This pattern was repeated for gel-free volume with stallions aged over twenty years recording 47% and 59% reduced counts compared to 5–9-year-old and 10–14-year-old stallions, respectively (*p* < 0.004).

### 3.3. Difference in Semen Characteristics among Stallion Disciplines

Kruskal–Wallis analyses found that significant differences (*p* < 0.05) occurred for all semen characteristics across the stallion disciplines investigated (Table 3). Post-hoc tests revealed multiple significant differences in semen characteristics among stallions in different disciplines (*p* < 0.008: Table 4). Stallions competing in show jumping had higher total and gel-free semen volumes than non-competing stallions (21% increase in TV; 24% increase in GFV, *p* < 0.0001) and dressage stallions (11% increase in TV; 13% increase in GFV, *p* < 0.0001). However, show jumping stallions recorded consistently lower values than dressage, eventing and non-competing stallions for sperm concentration (16% less than dressage; 23% less than eventing; 62% less than non-competing, *p* < 0.0001) and sperm progressive motility (8% less than eventing; 30% less than non-competing, *p* < 0.0001). Interestingly, non-competing stallions recorded significantly higher values than all other disciplines for sperm progressive motility (41% increase to show jumping; 38% increase to dressage; 30% increase to eventing, *p* < 0.0001), sperm concentration (163% increase to show jumping; 120% increase to dressage; 102% increase to eventing, *p* < 0.0001), total sperm count (127% increase to show jumping; 121% increase to dressage; 138% increase to eventing, *p* < 0.0001) and total progressively motile sperm (218% increase to show jumping; 206% increase to dressage; 212% increase to eventing, *p* < 0.0001).

### 3.4. Difference in Semen Characteristics among Stallion Competition Levels

The semen characteristics recorded varied with stallion competition level (Table 5). Significant differences (*p* < 0.0001) in all semen characteristics were found across the stallion competition levels investigated (Table 6), with the exception of total volume and gel-free volume which did not differ between non-competing and lower levels of competition, and total volume and gel-free volume which did not differ between non-competing and higher levels of competition. No significant differences were found in semen characteristics between lower and higher levels of competition. In addition, sperm progressive motility and sperm concentration did not differ between lower and elite levels of competition. No differences occurred in sperm progressive motility and sperm concentration between higher and elite levels.

Stallions competing at elite level of competition had significantly higher total and gel-free volumes of semen than non-competing stallions (16% increase in TV; 17% increase in GFV, *p* < 0.0001). Non-competing stallions had higher values for sperm progressive motility (26% higher than lower levels; 29% higher than higher levels; 26% higher than elite levels, *p* < 0.0001), sperm concentration (56% higher than lower levels; 53% higher than higher levels; 57% higher than elite levels, *p* < 0.0001), total sperm count (59% higher than lower levels; 58% higher than higher levels; 53% higher than elite levels, *p* < 0.0001) and total progressively motile sperm (70% higher than lower levels; 70% higher than higher levels; 65% higher than elite levels, *p* < 0.0001).

#### 3.4.1. Model A1: Effect of 18 Factors on total Semen Volume

Model A1 was performed to ascertain the effects of 18 independent variables (Appendix A) on the 6likelihood that total semen volume was above industry standards (Table 7). The model explained 88.3% (Nagelkerke R^2^) of the variance in total volume and correctly classified 95.9% of cases. ROC curve analysis indicated that the accuracy of Model A1 was excellent (0.987). Only five variables made a unique statistically significant contribution to the model. Show jumping stallions were 2.95:1 (*p* = 0.03) more likely to exhibit total semen volume above industry standards than any other discipline, however discipline itself did not prove to be a significant influencer (*p* > 0.05). Samples which contained an increased gel-free volume (>40 mL) were 1.25:1 mL (*p* = 0.0001) more likely to concurrently have a total semen volume above the industry standard. Similarly, increases in total progressively motile sperm (>2.4 × 10^9^) were associated with a 1.67:1 × 10^9^ increase (*p* = 0.0003) in total semen volume above the industry standard. Increases in sperm concentration (>450 × 10^6^) were associated with slightly reduced (0.98:1 × 10^6^) likelihood of above industry standard total semen volume, whilst increased sperm progressive motility was associated with a 0.94:1% (*p* = 0.001) decreased likelihood of showing total semen volume above industry standards. However, within this model, many of the confidence intervals (95% CI) presented values above 1, suggesting high variability in the data. Therefore, caution must be applied when interpreting the outcomes.

#### 3.4.2. Model A2–A5: Effect of 18 Factors on Gel-Free Volume, Spermatozoa Progressive Motility, Semen Concentration and Total Sperm Count

Model A2–A5 were performed to ascertain the effects of 18 independent variables (Appendix A) on the likelihood that gel-free semen volume (MA2), sperm progressive motility (MA3), sperm concentration (MA4) and total sperm count (MA5) were above industry standards. The models explained 100% (Nagelkerke R^2^) of the variance in the dependant variables and correctly classified 100% of cases. No differences were found in the odds of having above or below industry values for any variable evaluated in: Model A2, gel-free volume; Model A3, sperm progressive motility; Model A4, sperm concentration or Model A5, total sperm count (*p* > 0.05).

#### 3.4.3. Model A6: Effect of 18 Factors on Total Progressively Motile Sperm Count

Model A6 was performed to ascertain the effects of 18 independent variables (Appendix A) on the likelihood that total progressively motile sperm was above industry standards (Table 7). The model explained 82.7% (Nagelkerke R^2^) of the variance in total progressively motile sperm and correctly classified 98.1% of cases. A ROC curve analysis showed that Model A6 had excellent predictability (0.988). Three variables made a unique statistically significant contribution to the model. Increases in total sperm count (>6 × 10^9^) were 6.96:1 × 10^9^ (*p* = 0.0001) more likely to have total progressively motile sperm above the industry standard. Increases in gel-free semen volume (>40 mL) and sperm progressive motility (>40%) were associated with increased (1.02:1 mL; *p* = 0.038 and 1.25:1%; *p* = 0.0009, respectively) likelihood of above industry standards for total progressively motile sperm.

#### 3.4.4. Model B1: Effect of 15 Factors on Total Semen Volume

Model B1 was performed to ascertain the effects of 15 independent variables (Appendix A) on the likelihood that total semen volume was above industry standards (Table 7). The model explained 83.6% (Nagelkerke R^2^) of the variance in total volume and correctly classified 96.4% of cases. ROC curve analysis showed the predictability of Model B1 to be excellent (0.998). Two variables made individual statistically significant contributions to the model. Increases in total sperm count (>6 × 10^9^) were associated with being 3.91:1 × 10^9^ (*p* = 0.0006) more likely to concurrently record total semen volumes above industry standards, whilst sperm concentration was associated with a 0.94:1 × 10^6^ decreased (*p* = 0.0001) likelihood of displaying above industry values for total semen volume.

#### 3.4.5. Model B2–B5: Effect of 15 Factors on Gel-Free Volume, Spermatozoa Progressive Motility, Semen Concentration and Total Sperm Count

Model B2–B5 were performed to establish the effects of 15 independent variables (Appendix A) on the likelihood that gel-free semen volume (MB2), sperm progressive motility (MB3), sperm concentration (MB4) and total sperm count (MB5) were above industry standards. The models explained 100% (Nagelkerke R^2^) of the variance in the dependant variables and correctly classified 100% of cases. No differences were found in the odds of having above or below industry values for any variable evaluated in: Model B2, gel-free volume; Model B3, sperm progressive motility; Model B4, sperm concentration or Model B5, total sperm count (*p* > 0.05).

#### 3.4.6. Model B6: Effect of 15 Factors on Total Progressively Motile Sperm Count

Model B6 was performed to ascertain the effects of 15 independent variables (Appendix A) on the likelihood that total semen volume was above industry standards (Table 7). The model explained 87.9% (Nagelkerke R^2^) of the variance in total volume and correctly classified 99.4% of cases. ROC analysis indicated the predictability for Model B6 was excellent (0.989). Two variables made statistically significant contributions to the model. Increases in total sperm count (>6 × 10^9^) were 9.95:1 × 10^9^ (*p* = 0.0001) more likely to record total progressively motile sperm above industry standard. Increases in sperm progressive motility (>40%) were associated with a 1.23:1% increase (*p* = 0.0002) in total progressively motile sperm above industry standards.

Within all models many of the confidence intervals (95% CI) presented values above 1, suggesting high variability in the data, therefore caution must be applied when interpreting the outcomes.

## 4. Discussion

### 4.1. Semen Characteristics and Age

Age was found to significantly influence the quality of the seminal characteristics investigated, with a marked improvement in semen quality (sperm progressive motility, sperm concentration, sperm count and total progressively motile sperm) associated with stallions aged 20 years and above compared to all other age categories. These results differ from previous research within commercial breeding stallions [16,25,26], where declines in semen quality by the age of 10 years are consistently reported. This decline is postulated to be due to aging stallions being more susceptible to substandard spermatogenesis and testicular degeneration [16,26]. Whilst previous research states that stallion fertility is optimal at 5–9 years [15,26], the current study observed an increase of 23% in the total progressively motile sperm of stallions aged ≥20 years when compared to stallions aged 5–9 years. Within the 5–9 years category, 75% of stallions were competing within various disciplines, 67% of which were at higher or elite competition levels. Therefore, the declines observed here in semen quality could be an indicator that as competition performance peaks, reproductive performance is negatively affected. This could be due to the increase in workload required for optimum performance.

### 4.2. Difference in Semen Characteristics among Stallion Disciplines

To the authors’ knowledge, no previous equine studies have evaluated the effect of stallion competitive discipline participation on the quality of semen characteristics.

Within this study, competing stallions had significantly reduced semen characteristic values compared to non-competing stallions. This was seen for sperm progressive motility, sperm concentration, total sperm count and total progressively motile sperm. We suggest that the lower values observed within these semen characteristics may be related to disturbances within the endocrine system associated with exercise, leading to consequent disruption of spermatogenesis [27,28]. This could also explain why disciplines anecdotally considered less physiologically demanding had higher semen values. Dressage stallions had significantly higher sperm progressive motility, sperm concentration, total sperm count and total progressively motile sperm measurements compared to show jumping stallions. The resting heart rate of a horse is 32–36 bpm, which increases dependent upon type of exercise. Horses are shown to have reduced heart rates (46 bpm vs. 81.9 bpm) and cortisol levels (3.5 nmol/L vs. 5.01 nmol/L) when comparing dressage and show jumping stallions, respectively [29,30], most likely reflecting differences in discipline workload related to exercise frequency, duration and intensity [30]. Therefore, it could be postulated that dressage competition has a reduced impact on stallion endocrinology, due to the lower competition intensities and energy demands [29].

The thermal effect of higher intensity exercise could also impact on semen characteristics [31]. Interestingly, a reduced total sperm count was found in eventing horses compared to other disciplines. Following exercise, stallion internal scrotal temperature has been recorded to rise over 2 °C [32]. This can cause testicular hyperthermia and thermal insult could negatively impact on spermatogenesis [31,32,33]. Whilst thermal stress could explain some of the negative effects seen in semen characteristics among stallion disciplines, this is still a debated topic area within equine research [10,31,32,34]; and no definitive conclusions have been drawn. Further studies are required to investigate this effect and would benefit from the inclusion of stallions within the competitive environment and the classification of exercise intensity, as well as measuring both subcutaneous testicular and rectal temperatures concurrently with the evaluation of semen characteristics.

Significantly higher semen volumes were found within competing stallions, with the lowest volumes within non-competing stallions. This is thought to be due to increases in prolactin as a result of exercise [35,36]. Fertility is not considered to be directly affected by semen volume [37]; therefore, the lower values seen within non-competing stallions should not be considered a negative impact on semen quality, unless they fall below industry standards or what would be classed as “normal” for individual stallions.

### 4.3. Difference in Semen Characteristics among Stallion Competition Levels

Elite stallions were found to have significantly higher values in total sperm count and total progressively motile sperm compared to all other competition levels for the actively competing stallions. This does not concur with findings from previous studies [4,14,28], which found that higher intensity exercise have detrimental effects on semen characteristics. Horses competing at elite levels are managed for the highest level of athletic performance, which may equate to less demanding training regimes and stringently managed competition schedules. This may not occur with stallions at lower levels of competition. This optimised management may attenuate the negative impact of exercise on semen quality within the elite level stallions. Future research should explore the management practices of competing horses and breeding stallions with the goal of creating a successful management protocol for dual career stallions to enhance performance and breeding.

Interestingly, no significant differences were observed between lower and higher competition level horses for any of the semen characteristics. It has been observed that horses competing within non-elite level dressage train for a duration of approximately one hour, three to four times per week, suggesting similar exercise regimes [38]. The results seen within this study could be resultant of similar management protocol between the two competition levels.

In the present study, non-competing stallion sperm concentration, total sperm count and total progressively motile sperm values were increased compared to competing stallions. These increases may be associated with differences in individual horses’ management regimes [39,40,41]. Therefore, whilst it may be possible to optimise the management of dual career stallions for competition performance, it may be difficult to also improve breeding performance concurrently. Previous research states that disruptions within spermatogenesis can take up to 57 days for semen characteristics to return to basal values [40]. Our results suggest that both competition level and discipline have significant negative effects on semen characteristics and we would recommend that competition programs for dual career stallions should be managed to ensure sufficient time in light-moderate exercise and that stallions be removed from active competition before collection takes place. Alternatively, it could be suggested that competition stallions should have semen collections scheduled outside of competition seasons and semen frozen for use within the consecutive competition season. Sperm evaluation for morphological defects could assist with detecting the time frame in which damage to the sperm occurs. Due to the retrospective nature of the data within the current study, this information was not available, however future work integrating sperm morphology data to gain a greater understanding on the stallions testicular functioning is warranted.

### 4.4. Competition Level and Discipline Association with Semen Characteristics Presenting as above or below Industry Standards

We found stallion competition level and discipline did not significantly affect whether the semen characteristics investigated were above or below the current industry standards for artificial insemination. Since differences among semen characteristics were found between competing level and discipline, this suggests that confounding management factors which were not recorded could be influential. Future prospective and longitudinal studies exploring the impact of various management factors, for example nutrition and number of competitions attended, are warranted to identify if further risk factors are associated with semen quality in competing stallions.

## 5. Conclusions

Competing stallions within dressage, show jumping and eventing had significantly lower quality semen than non-competing stallions. In contrast to previous research, stallions aged 20 years and over presented with increased semen characteristic values. This may be related to their non-competitive status, suggesting that competitive activity may have a greater negative impact on spermatogenesis than age. Stallions at elite level of competition recorded higher total sperm count and sperm progressive motility compared to stallions active within lower competition levels. This suggests that the management for optimal performance may attenuate the negative impacts on semen quality which can occur with exercise. 

## Figures and Tables

**Figure 1 animals-09-00485-f001:**
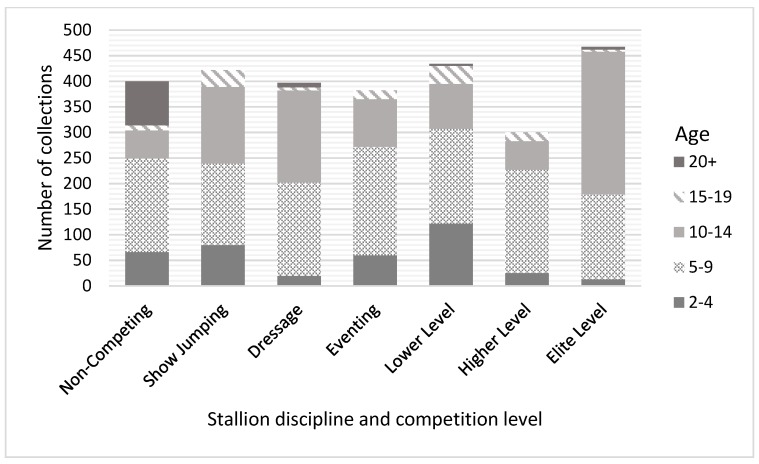
Number of semen collections from each age category within stallion discipline and competition level.

**Table 1 animals-09-00485-t001:** Medians, ranges and interquartile ranges for each semen characteristic among age categories of competing and non-competing stallions.

Semen Characteristic	Median					Range					Interquartile Range				
	2–4	5–9	10–14	15–19	20+	2–4	5–9	10–14	15–19	20+	2–4	5–9	10–14	15–19	20+
	(*n* = 226)	(*n* = 736)	(*n* = 478)	(*n* = 67)	(*n* = 94)	(*n* = 226)	(*n* = 736)	(*n* = 478)	(*n* = 67)	(*n* = 94)	(*n* = 226)	(*n* = 736)	(*n* = 478)	(*n* = 67)	(*n* = 94)
Total volume (mL)	42.00	54.00	59.00	49.00	44.00	1–120	17–290	15–220	17–198	25–103	28.00	36.00	35.00	43.00	22.00
Gel-free volume (mL)	34.00	42.00	48.00	36.50	30.00	3–110	5–150	3–180	5–185	29,495.00	24.00	34.00	34.00	44.00	20.00
Progressive motility (%)	65.00	70.00	65.00	70.00	90.00	34,820.00	0–95	25–95	40–85	50–95	20.00	15.00	15.00	21.00	10.00
Concentration (×10^6^/mL)	263.00	243.00	199.00	284.00	434.00	1–852	11–789	20–691	71–611	71–846	229.00	246.00	160.00	258.00	155.00
Total sperm count (×10^9^)	9.04	9.46	9.54	10.22	12.67	0.03–40.85	0.44–89.25	031–28.99	1.33–39	1.42–38.32	7.69	7.92	5.75	7.58	10.67
Total progressively motile sperm (×10^9^)	5.84	5.99	5.99	6.30	10.82	0.01–36.77	0–75.86	0.15–23.61	0.99–25.35	0.78–34.49	5.90	5.97	4.15	3.81	8.47

**Table 2 animals-09-00485-t002:** Results (probability values) of pairwise comparisons to identify where differences in semen characteristics occur within stallion age categories.

Stallion Age Range (years)	2–4	5–9	10–14	15–19	20+
2–4		TV = 0.0001 *	TV = 0.0001 *	TV = 0.016	TV = 0.936
		GFV = 0.0001 *	GFV = 0.0001 *	GFV = 0.224	GFV = 0.046
		PM = 0.928	PM = 0.021	PM = 0.037	PM = 0.0001 *
		Conc = 0.009	Conc = 0.0001 *	Conc = 0.601	Conc = 0.0001 *
		TSC = 0.320	TSC = 0.493	TSC = 0.280	TSC = 00001 *
		TPMS = 0.409	TPMS = 0.930	TPMS = 0.788	TPMS = 0.0001 *
5–9			TV = 0.011	TV = 0.410	TV = 0.0001 *
			GFV = 0.004 *	GFV = 0.324	GFV = 0.0001 *
			PM = 0.0001 *	PM = 0.033	PM = 0.0001 *
			Conc = 0.0001 *	Conc = 0.384	Conc = 0.0001 *
			TSC = 0.870	TSC = 0.565	TSC = 0.0001 *
			TPMS = 0.468	TPMS = 0.933	TPMS = 0.0001 *
10–14				TV = 0.096	TV = 0.0001 *
				GFV = 0.048	GFV = 0.0001 *
				PM = 0.665	PM = 0.0001 *
				Conc = 0.002 *	Conc = 0.0001 *
				TSC = 0.385	TSC = 0.0001 *
				TPMS = 0.808	TPMS = 0.0001 *
15–19					TV = 0.045
					GFV = 0.037
					PM = 0.0001 *
					Conc = 0.0001 *
					TSC = 0.010
					TPMS = 0.0001 *

TV, total volume; GFV, gel-free volume; PM, progressive motility; Conc, concentration; TSC, total sperm count; TPMS, total progressively motile sperm; * denotes significant result (Bonferroni adjusted alpha *p* < 0.005).

**Table 3 animals-09-00485-t003:** Medians, ranges and interquartile ranges for each semen characteristic among varying stallion disciplines.

Semen Characteristic	Median				Range				Interquartile Range			
	Non-Comp	SJ	Dress	Event	Non-Comp	SJ	Dress	Event	Non-Comp	SJ	Dress	Event
	(*n* = 400)	(*n* = 422)	(*n* = 397)	(*n* = 382)	(*n* = 400)	(*n* = 422)	(*n* = 397)	(*n* = 382)	(*n* = 400)	(*n* = 422)	(*n* = 397)	(*n* = 382)
Total volume (mL)	47.00	64.00	57.00	48.00	19–187	1–290	17–185	15–130	29.00	38.00	32.00	28.00
Gel-free volume (mL)	35.00	52.00	44.00	35.00	6–150	3–185	5–145	3–115	25.00	34.00	30.00	29.00
Progressive motility (%)	85.00	62.50	65.00	70.00	65–95	5–80	5–80	0–85	10.00	15.00	10.00	10.00
Concentration (×10^6^/mL)	470.50	165.00	200.00	225.00	301–852	1–550	42–632	11–543	143.00	114.00	137.00	137.00
Total sperm count (×10^9^)	16.11	8.13	8.60	7.56	2.39–89.86	0.03–39.00	1.03–27.90	0.44–28.40	12.74	5.47	4.86	5.71
Total progressively motile sperm (×10^9^)	13.85	5.06	5.53	5.03	2.03–75.86	0.01–25.35	0.41–18.14	0–20.04	11.50	3.40	3.37	3.71

Non-comp, non-competing; SJ, show jumping; Dress, dressage; Event, eventing.

**Table 4 animals-09-00485-t004:** Pairwise comparisons to identify where differences in semen characteristics occur within stallion disciplines.

Stallion Discipline	Non-Competing	Show Jumping	Dressage	Eventing
Non-competing		TV = 0.0001 *	TV = 0.0001 *	TV = 0.495
		GFV = 0.0001 *	GFV = 0.0001 *	GFV = 0.427
		PM = 0.0001 *	PM = 0.0001 *	PM = 0.0001 *
		Conc = 0.0001 *	Conc = 0.0001 *	Conc = 0.0001 *
		TSC = 0.0001 *	TSC = 0.0001 *	TSC = 0.0001 *
		TPMS = 0.0001 *	TPMS = 0.0001 *	TPMS = 0.0001 *
Show jumping			TV = 0.0001 *	TV = 0.0001 *
			GFV = 0.0001 *	GFV = 0.0001 *
			PM = 0.036	PM = 0.0001 *
			Conc = 0.0001 *	Conc = 0.0001 *
			TSC = 0.299	TSC = 0.062
			TPMS = 0.079	TPMS = 0.790
Dressage				TV = 0.0001 *
				GFV = 0.0001 *
				PM = 0.0001 *
				Conc = 0.001 *
				TSC = 0.006 *
				TPMS = 0.180

TV, total volume; GFV, gel-free volume; PM, progressive motility; Conc, concentration; TSC, total sperm count; TPMS, total progressively motile sperm; * denotes significant result (Bonferroni adjusted alpha *p* < 0.008).

**Table 5 animals-09-00485-t005:** Medians, ranges and interquartile ranges for each semen characteristic among stallion competition levels.

Semen Characteristic	Median				Range				Interquartile Range			
	Non-Comp	Lower	Higher	Elite	Non-Comp	Lower	Higher	Elite	Non-Comp	Lower	Higher	Elite
	(*n* = 400)	(*n* = 434)	(*n* = 300)	(*n* = 467)	(*n* = 400)	(*n* = 434)	(*n* = 300)	(*n* = 467)	(*n* = 400)	(*n* = 434)	(*n* = 300)	(*n* = 467)
Total volume (mL)	47.00	53.00	49.00	59.00	19–187	1–220	16–185	15–290	29.00	40.00	33.00	30.00
Gel-free volume (mL)	35.00	42.00	38.00	48.00	6–150	5–185	3–145	2–135	25.00	39.00	33.00	27.00
Progressive motility (%)	85.00	65.00	65.00	65.00	65–95	5–80	0–85	10–85	10.00	10.00	15.00	10.00
Concentration (×10^6^/mL)	470.50	195.50	210.50	188.00	301–852	1–628	11–621	20–550	143.00	151.00	162.00	107.00
Total sperm count (×10^9^)	16.11	7.40	8.14	9.02	2.39–89.86	0.03–28.40	0.38–225.55	0.47–39.00	12.74	5.31	5.24	4.84
Total progressively motile sperm (×10^9^)	13.85	4.87	4.87	5.70	2.03–75.86	0.001–18.88	0.00–15.33	0.16–25.35	11.50	3.55	3.40	3.31

Non-comp, non-competing.

**Table 6 animals-09-00485-t006:** Pairwise comparisons to identify where differences in semen characteristics occur among stallion competition levels.

Stallion Competition Level	Non-Competing	Lower Level	Higher Level	Elite Level
Non-competing		TV = 0.190	TV = 0.063	TV = 0.0001 *
		GFV = 0.030	GFV = 0.041	GFV = 0.0001 *
		PM = 0.0001 *	PM = 0.0001 *	PM = 0.0001 *
		Conc = 0.0001 *	Conc = 0.0001 *	Conc = 0.0001 *
		TSC = 0.0001 *	TSC = 0.0001 *	TSC = 0.0001 *
		TPMS = 0.0001 *	TPMS = 0.0001 *	TPMS = 0.0001 *
Lower level			TV = 0.570	TV = 0.0001 *
			GFV = 0.335	GFV = 0.002 *
			PM = 0.100	PM = 0.597
			Conc = 0.082	Conc = 0.895
			TSC = 0.245	TSC = 0.0001 *
			TPMS = 0.869	TPMS = 0.0001 *
Higher level				TV = 0.0001 *
				GFV = 0.0001 *
				PM = 0.033
				Conc = 0.056
				TSC = 0.001 *
				TPMS = 0.0001 *

TV, total volume; GFV, gel-free volume; PM, progressive motility; Conc, concentration; TSC, total sperm count; TPMS, total progressively motile sperm; * denotes significant result (Bonferroni adjusted alpha *p* < 0.008).

**Table 7 animals-09-00485-t007:** Multivariate model variables that were significant when tested against the dichotomous outcome of being above or below the industry standard measure for each of the semen characteristics measured in competing (*n* = 400 semen collections) and non-competitive (*n* = 1201 semen collections: show jumping, *n* = 422; dressage, *n* = 397; eventing, *n* = 382) warmblood sports horse stallions, collected from 2009 to 2016.

Model/Dependant Variable	Significant Variables	B	SE	*p*-Values	E × P(B)	95% Cl (L:U)
A1/total volume	Show jumping	0.952	0.447	0.033	2.591	1.079:6218
A1/total volume	Gel-free volume	0.221	0.026	0.01 × 10^−15^	1.247	1.185:1.312
A1/total volume	Progressive motility	−0.065	0.020	0.001	0.937	06.901:0.975
A1/total volume	Concentration	−0.016	0.005	0.003	0.984	0.974:0.995
A1/total volume	Total progressively motile sperm	0.511	0.143	0.0003	1.667	1.260:2.207
A6/total progressively motile sperm	Gel-free volume	0.023	0.011	0.038	1.024	1.001:1.046
A6/total progressively motile sperm	Progressive motility	0.225	0.028	0.009 × 10^−13^	1.252	1.185:1.323
A6/total progressively motile sperm	Total sperm count	1.940	0.199	0.01 × 10^−20^	6.960	4.714:10.275
B1/total volume	Concentration	−0.066	0.007	0.01 × 10^−18^	0.936	0.923:0.949
B1/total volume	Total sperm count	1.363	0.145	0.006 × 10^−18^	3.908	2.940:5.194
B6/total progressively motile sperm	Progressive motility	0.204	0.049	0.029 × 10^−3^	1.226	1.114:1.349
B6/total progressively motile sperm	Total sperm count	0.097	0.405	0.014 × 10^−10^	9.947	4.499:21.990

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
