# Peer review of "Variance in Stallion Semen Quality among Equestrian Sporting Disciplines and Competition Levels"

_animals, 2019, doi:10.3390/ani9080485_

Round 1

Reviewer 1 Report

The authors aimed to evaluate the influence of different disciplines and competition levels on horse semen parameters normally assessed during routine andrological evaluation in the field.

There are very interesting outcomes from data analysis and I believe that this manuscript provides good scientific information that can further contribute to more detailed studies in the reproduction of competition horses.

My main concerns are related to the lack of more specific information regarding exercise routine/training that was not recorded, as well as the interval between semen collections. According to the authors, “confounding managements factors which were not recorded could be influential”.  Therefore, part of the discussion is very speculative and should be improved. It is also very important to record and insert in the manuscript data about the interval between semen collections, as this variable can influence semen characteristics.

L29 – Insert a dot before “Research”

L36 – Insert a dot before “Studies”

L49 – I believe the statement is missing information.

L53 – Competition levels – not all the articles cited (15, 16, 17, and 18) describe clearly and similarly the topic. Please, insert objectively how this variable was scored. Also, insert the intervals between semen collections.

- Data Analysis – Why there is no information about sperm morphology? In L344 the authors justify that morphologic characteristics were not assessed due to “retrospective nature of the data”.  However, sperm morphology analysis is part of the “Guide to the use of artificial insemination in the horse breeding”. Is it not possible to assess this data? I believe that it could elucidate more clearly the influence of disciplines and competition level on semen quality.

- Figure 1 – Why the age of younger horses were 1-4, unlike further Tables (2-4)?

L128 – Define the acronyms before first use.

L177 – Table 6, not Table 5.

L312 – “the lower values seen within competing stallions”. As the authors are talking about semen volume and considering the results presented, I believe they mean “the higher values seen within competing stallions”.

Table 1 – The title has to be clearer about data origin (sports and non-competing horses, the period of semen collection, warmbloods horses). Besides, the Table formatting needs to be adjusted. The total number of semen collections presented is 1,600 (226+736+478+66+94=1,600), differing from M & M. Standardization is also necessary regarding decimal numbers.

- Tables 1,3, and 5 – How total volume range can start from zero? If there was no ejaculate, how this data was included? Moreover, even when the total volume range starts from zero, gel-free volume range starts from higher values (Table 1: 3; Table 3:3; Table 5:5). What is the explanation for that?

- In my opinion, the authors should merge data from both Table 1 and 2. Display means/median and dispersion measure, range and probability for each parameter evaluated (according to age). The same for Tables 3 and 4, 5 and 6.

- Use “progressive motility” or “progressively motile” instead “progressively motility” throughout the manuscript.

- Table 7 - The title has to be clearer about data origin (sports and non-competing horses, the period of semen collection, warmbloods horses, sample number)

The paper below could contribute to discussion:

https://doi.org/10.15232/S1080-7446(15)30269-2

Author Response

Dear Editor and anonymous reviewers,

We have revised our submission “Variance in stallion semen quality between equestrian sporting disciplines and competition levels” in response to the review. We would like to thank the reviewers for their comments which have helped substantially to improve the manuscript. We would like to also thank the editor for considering this revised submission.

Whilst amendments have of course been made in the manuscript as per the observations, please find detailed below our responses to the editors and reviewers’ comments:

Reviewer 1

Comments and Suggestions for Authors

The authors aimed to evaluate the influence of different disciplines and competition levels on horse semen parameters normally assessed during routine andrological evaluation in the field.

There are very interesting outcomes from data analysis and I believe that this manuscript provides good scientific information that can further contribute to more detailed studies in the reproduction of competition horses.

Thank you

 My main concerns are related to the lack of more specific information regarding exercise routine/training that was not recorded, as well as the interval between semen collections. According to the authors, “confounding managements factors which were not recorded could be influential”.  Therefore, part of the discussion is very speculative and should be improved. It is also very important to record and insert in the manuscript data about the interval between semen collections, as this variable can influence semen characteristics.

Thank you for your kind words and constructive criticism, which has helped us improve on our manuscript

L29 – Insert a dot before “Research” =This has been rectified.

L36 – Insert a dot before “Studies” =This has been rectified.

L49 – I believe the statement is missing information. =Inserted that competition level was also investigated.

L53 – Competition levels – not all the articles cited (15, 16, 17, and 18) describe clearly and similarly the topic. Please, insert objectively how this variable was scored. Also, insert the intervals between semen collections. =Information on competition level eligibility have now been stated.

- Data Analysis – Why there is no information about sperm morphology? In L344 the authors justify that morphologic characteristics were not assessed due to “retrospective nature of the data”.  However, sperm morphology analysis is part of the “Guide to the use of artificial insemination in the horse breeding”. Is it not possible to assess this data? I believe that it could elucidate more clearly the influence of disciplines and competition level on semen quality. =Unfortunately, due to the retrospective nature of the data I did not have access to morphology information. The data was received from working stud farms which stated themselves that unless a major issue occurred in a stallion, morphology was never assessed due to time constraints. 

- Figure 1 – Why the age of younger horses were 1-4, unlike further Tables (2-4)? =This has been rectified.

L128 – Define the acronyms before first use. =This has been corrected throughout.

L177 – Table 6, not Table 5. =This has been corrected.

L312 – “the lower values seen within competing stallions”. As the authors are talking about semen volume and considering the results presented, I believe they mean “the higher values seen within competing stallions”. =This has been amended.

Table 1 – The title has to be clearer about data origin (sports and non-competing horses, the period of semen collection, warmbloods horses). Besides, the Table formatting needs to be adjusted. The total number of semen collections presented is 1,600 (226+736+478+66+94=1,600), differing from M & M. Standardization is also necessary regarding decimal numbers. =This has been rectified.

- Tables 1,3, and 5 – How total volume range can start from zero? If there was no ejaculate, how this data was included? Moreover, even when the total volume range starts from zero, gel-free volume range starts from higher values (Table 1: 3; Table 3:3; Table 5:5). What is the explanation for that?

- In my opinion, the authors should merge data from both Table 1 and 2. Display means/median and dispersion measure, range and probability for each parameter evaluated (according to age). The same for Tables 3 and 4, 5 and 6. =We tested this on colleagues and the general consensus was the tables were easier to interpret separately, therefore the tables have not be merged for clarity of results.

- Use “progressive motility” or “progressively motile” instead “progressively motility” throughout the manuscript. =This has been altered throughout.

- Table 7 - The title has to be clearer about data origin (sports and non-competing horses, the period of semen collection, warmbloods horses, sample number) = amended to identify these horses are sports horses and to include sample number

Reviewer 2 Report

Your work represents a benefit in the field and hopefully will contribute to positive change in the management and use of the competition horses, both for their dual purpose and life quality.  Well written. Only very minor changes of wording may be needed.  For the supplementary files the only recommendation would be to move the legends below the tables.

L24: replace ‘competition’ with ‘competitions’

L24: insert ‘that’ between ‘mean’ and ‘successful’

L35: citation needed after ‘humans’

L36: insert a dot after [13]

L38: insert an apostrophe after stallions (stallions’ semen quality)

L42: insert citation after ‘characteristics’ or delete ‘competing’

L49: for the final version complete this sentence

L124: remove ‘existed’

L145: Table 1: Please arrange this table and the following ones, where it applies, to fit properly in the page. Use lower case in the second word of heading titles (i.e. Semen characteristic, Total volume, Gel-free volume and so on). Do not split the numbers in consecutive rows. Remove the free rows within the table or the spacing after the rows. Do not color the background in the cells. Remove the horizontal lines except the ones at the beginning of the table, after the heading cells and at the end of the table. Separate the columns properly in the headings.

Change ‘Progressively Motility’ to ‘Progressive motility’

L147: Table 2: Please move the legend below this table and the following ones where it applies. Change the headings to be more explanatory for the readers. Remove the background colors from the cells.

L167: The same recommendations as for Table 1 and Table 2

Change ‘Progressively Motility’ to ‘Progressive motility’

L170: Please see the recommendations above, as for the other tables

L192: The same recommendations as for Table 1 and Table 2

L194:  The same recommendations as for Table 1 and Table 2

L202: insert ‘that’ between ‘indicated’ and ‘the accuracy’

L215: The same recommendations as for Table 1 and Table 2

L272: change ‘were’ to ‘where’

L318: change ‘who found’ to ‘which found that’

L319: change ‘had’ to ‘have’

L320: remove ‘of competition’

L340: change ‘are’ to ‘to be’

L341-343: Nothing to change, just as observation: in English Thoroughbred breeding this would be difficult due the tradition for natural covering.

Author Response

Comments and Suggestions for Authors

Your work represents a benefit in the field and hopefully will contribute to positive change in the management and use of the competition horses, both for their dual purpose and life quality.  Well written. Only very minor changes of wording may be needed.  For the supplementary files the only recommendation would be to move the legends below the tables.

 Thank you for your kind words and helpful comments.

L24: replace ‘competition’ with ‘competitions’ =This has been rectified.

L24: insert ‘that’ between ‘mean’ and ‘successful’ =This has been corrected.

L35: citation needed after ‘humans’ =Citation has been added

L36: insert a dot after [13] =This has been done.

L38: insert an apostrophe after stallions (stallions’ semen quality) =This has been done.

L42: insert citation after ‘characteristics’ or delete ‘competing’ =This has been altered.

L49: for the final version complete this sentence =Not sure what this comment is referring to.

L124: remove ‘existed’ =This has been done.

L145: Table 1: Please arrange this table and the following ones, where it applies, to fit properly in the page. Use lower case in the second word of heading titles (i.e. Semen characteristic, Total volume, Gel-free volume and so on). Do not split the numbers in consecutive rows. Remove the free rows within the table or the spacing after the rows. Do not color the background in the cells. Remove the horizontal lines except the ones at the beginning of the table, after the heading cells and at the end of the table. Separate the columns properly in the headings. =The tables throughout have been altered in accordance with this reviewers comments.

Change ‘Progressively Motility’ to ‘Progressive motility’ =This has been changed.

L147: Table 2: Please move the legend below this table and the following ones where it applies. Change the headings to be more explanatory for the readers. Remove the background colors from the cells. =This has been rectified.

L167: The same recommendations as for Table 1 and Table 2 =The tables throughout have been altered in accordance with this reviewers comments.

Change ‘Progressively Motility’ to ‘Progressive motility’ =This has been rectified.

L170: Please see the recommendations above, as for the other tables =The tables throughout have been altered in accordance with this reviewers comments.

L192: The same recommendations as for Table 1 and Table 2 =The tables throughout have been altered in accordance with this reviewers comments.

L194:  The same recommendations as for Table 1 and Table 2 =The tables throughout have been altered in accordance with this reviewers comments.

L202: insert ‘that’ between ‘indicated’ and ‘the accuracy’ =This has been done.

L215: The same recommendations as for Table 1 and Table 2 =The tables throughout have been altered in accordance with this reviewers comments.

L272: change ‘were’ to ‘where’ =This has been changed.

L318: change ‘who found’ to ‘which found that’ =This has been altered.

L319: change ‘had’ to ‘have’ =This has been changed.

L320: remove ‘of competition’ =This has been removed.

L340: change ‘are’ to ‘to be’ =This has been changed.

L341-343: Nothing to change, just as observation: in English Thoroughbred breeding this would be difficult due the tradition for natural covering.